# Exploring Gut Microbiome Variations between *Popillia japonica* Populations of Azores

**DOI:** 10.3390/microorganisms11081972

**Published:** 2023-07-31

**Authors:** Jorge Frias, Anna Garriga, Ángel Peñalver, Mário Teixeira, Rubén Beltrí, Duarte Toubarro, Nelson Simões

**Affiliations:** 1Centro de Biotecnologia dos Açores, Faculdade de Ciências e Tecnologia, Universidade dos Açores, 9500-321 Ponta Delgada, Portugal; 2Departament de Biologia Animal, Vegetal i Ecologia, Facultat de Biociències, Universitat Autònoma de Barcelona, 08193 Bellaterra, Spain

**Keywords:** *Popillia japonica*, associated microbiome, bacterial diversity, enriched phyla, enriched pathways

## Abstract

*Popillia japonica* (*Coleoptera*: *Scarabaeidae*), is an emerging invasive pest in Europe and America. In the Azores, this pest was first found on Terceira Island during the sixties and soon spread to other islands. The rate of infestation differs between islands, and we hypothesized that microbiome composition could play a role. Therefore, we sampled 3rd instar larvae and soil from sites with high and low infestation rates to analyze the microbiome using next-generation sequencing. We analyzed twenty-four 16S DNA libraries, which resulted in 3278 operational taxonomic units. The alpha and beta diversity of the soil microbiome was similar between sites. In contrast, the larvae from high-density sites presented a higher bacterial gut diversity than larvae from low-density sites, with biomarkers linked to plant digestion, nutrient acquisition, and detoxification. Consequently, larvae from high-density sites displayed several enriched molecular functions associated with the families *Ruminococcaceae*, *Clostridiaceae* and *Rikenellaceae*. These bacteria revealed a supportive function by producing several CAZyme families and other proteins. These findings suggest that the microbiome must be one drive for the increase in *P. japonica* populations, thus providing a checkpoint in the establishment and spread of this pest.

## 1. Introduction

*Popillia japonica* Newman (*Coleoptera*, *Scarabaeidae*) is a highly voracious herbivorous insect that consumes a wide variety of plants, many of which are of high economic value. *P. japonica* has been designated as one of 2019’s priority pests by the European Commission. Adults are attracted to foliage, flowers, and fruits and can cause significant damage to ornamental plants and fruit and vegetable crops. Larvae develop in the grass and can cause serious damage to turfgrasses and pastures. *P. japonica* is native to Japan; thus, it is commonly known as the Japanese beetle (JB). It became an invasive pest in the 20th century; it had arrived in the United States by 1916 [1] and later spread to other countries, where it is considered a quarantine pest. In the Azores, *P. japonica* was first noticed in the mid-1960s on Terceira Island near a military airport used by the US Air Force [2]. Without any natural enemies, the pest spread from the introduction point to the most favorable environments, particularly to permanent pastures with high humidity and mild temperatures [3] without any significant impact of lure traps applied on the most infested areas [4]. Moreover, no effective reduction in the population was observed, after the application of chemical pesticides against adults applied in a limited area for two consecutive years. The application of exogenous and local strains of entomopathogenic nematodes, despite their good infectivity, was not efficient in reducing the population, mostly because the establishment of these pathogens was not effective [5,6,7,8,9]. The same occurred with applied *Bacillus popilliae* strains [10]. As a consequence of failed control measures, the insect colonized the island quickly. To avoid spreading to the other islands of the archipelago, the regional government-imposed quarantine measures in 1985, restricting the exportation of any material that could transport adults or immature stages of *P. japonica*. Moreover, traps were placed in all ports and airports of the archipelago islands to serve as warning stations. Despite this, in 2013, the presence of the pest in another seven islands was reported by the Agricultural Services of Azores (J. Mota, personal communication, [11]), apparently, with different population rates.

This unequal distribution can be influenced by many factors such as vegetation, predators, pathogens, climatic variables, anthropogenic actions, etc. Microbial communities in the soil and associated with the insect could be factors driving the insect population. Aiming to understand the role of the microbiome in population rates we chose two islands with quite similar ecosystems but with different population rates to analyze and compare the microbiome of soils and gut larvae. The soil microbiome has an important impact on insects, particularly on soil-dwelling insects, that is the case of *P. japonica* whose immature stages spend more than 75% of its life cycle in the soil, feeding not only on roots but also on organic matter, as recently revealed [12], thus exposed to a huge variety of microorganisms [13,14]. A few studies have evidence that the soil microbiome can affect the insect immune system and resistance against soil pathogens by interfering with the insect microbiome [15].

Moreover, the gut microbiome of insects is known to have a great impact on their survival and expansion. Previous studies have demonstrated the fundamental importance of microbes in the adaptation of insect pests to new environments [16,17,18]. The gut microbiome assists in the breakdown of the thick cell walls of plant roots and the extraction of essential nutrients for the development and growth of the insect host. In addition to their role in nutrient acquisition, the microbiome also plays an important role in protecting JB grubs from pathogens [19,20]. According to Schloss et al., the immune system of insects is strengthened by the gut microbiome, which produces antimicrobial substances that compete with and even inhibit the growth of pathogens, including other bacteria and fungi. Moreover, the loss of microorganisms frequently causes improper growth and decreases host survival [21]. A variety of environmental factors, including food availability, soil type, and exposure to other species have been shown to affect the gut microbiome of JB grubs [22,23]. Furthermore, the gut microbiome of grubs from various populations, inhabiting various environments or consuming various plants, can drastically differ. Regarding different insect pests, the microorganisms present in the digestive system are known to influence the survival and development of the grubs, so far [19,24,25]. However, more investigation is required to comprehend how the gut microbiome contributes to the survival and dispersal of this pest around the globe and eventually to find key targets to use in the prevention of insect installation and growth. For this reason, we hypothesize that differences in the microbiome of soil and larvae in the two islands of the Azores could provide essential information on the influence of the microbiome on the success of a population. Therefore, the present work aims to answer the following questions: (1) Are there differences in the microbiomes of soils and gut of larvae between high- and low-density sites? (2) Does the microbiome of larvae vary according to the soil microbiome? (3) Do abiotic factors modulate the microbiome of soil and consequently the gut microbiome of grubs? For this, we collected samples of *P. japonica* grubs and soil from populations with low density (São Miguel Island) and from populations with high density (São Jorge Island) and analyzed the bacterial composition of these samples using next-generation sequencing (NGS) methods.

## 2. Materials and Methods

### 2.1. Collection and Processing of Insect and Soil Samples

For this study, we chose a site with high infestation levels of *P. japonica* which corresponded to São Jorge Island (HD) with an annual mean of 38,323 captured adults/km^2^ and a site with low infestation levels which corresponded to São Miguel Island (LD) with 1106 captured adults/km^2^, based on 10 years record of JB adults collected in the Azores supported by the Regional Secretariat for Agriculture and Rural Development of the Government of the Azores [26]. On each island, we delimited 40 km^2^ of study sites in which three stations with identical altitudes and vegetation among those lands used for pasture and with the highest known infestation rates per island (Appendix A). The selected pastures in both islands exhibited similar flora composition, primarily consisting of white clover (*Trifolium repens* L.), perennial ryegrass (*Lolium perenne* L.), and annual ryegrass (*Lolium multiflorum Lam.*) [27]. The pastures are surrounded mainly by *Rubus* spp. (blackberry, raspberry). Subsequently, we randomly chose two sampling spots from each station and collected a total of six soil and larvae samples per density between April and May 2021 (Appendix A).

We collected the soil samples (1 kg each) by mixing ten subsamples, kept them cold during transport to the laboratory, and finally stored them at −80 °C. Also, we collected six biological replicates of larvae, each consisting of a pool of five larvae samples from the same random spots where we collected the soil samples. The larvae samples were collected in refrigerated absolute ethanol and transported in a thermo-container at 4 °C. Approximately 16 h after collection, the larvae were surface-sterilized using the protocol described by Montagna et al. [18] and then aseptically dissected in sterile Ringer solution to obtain the whole gut.

From each soil sample, we performed five DNA extractions from 10 mg of soil which were gathered together to be sequenced as one representative sample. DNA was extracted from a gut homogenate and soil using a DNeasy Power Soil Kit (Qiagen, Hilden, Germany), following the manufacturer’s instructions, and preserved at −80 °C. The DNA quality and purity were assessed with a fluorometer (Qubit^TM^ 2.0, Singapure) and dsDNA BR Assay kit (Qubit^TM^, Eugene, OR, USA) with quantity thresholds of 73 and 270 ng/µL per sample, respectively.

### 2.2. Soil and Climatic Parameters

The climatic parameters (pluviometry, temperature, relative humidity, and radiation) were obtained from official stations located in Velas (São Jorge) and Ponta Delgada (São Miguel). The mean climatic values were calculated based on the annual period of larvae from January to May and from October to December, from 2020 to 2021.

The collected soil samples were analyzed at the Laboratory of Soil and Plant Analysis at the University of the Azores, with the following order of tasks: the soil was dried in an oven with forced air circulation at 40 °C for two to three days and sifted using a 2 mm mesh sieve to obtain fine soil.

Subsequently, chemical analysis of the following parameters was carried out: texture, using the densimeter method, according to Bouyoucos [28]; total organic matter, determined by calcination; and pH, determined potentiometrically in water with a glass electrode (10:25) (Hanna instruments, Lingolsheim, France).

The macronutrients potassium (K), magnesium (Mg), and calcium (Ca) were determined by extraction in ammonium acetate (NH_4_C_2_H_3_O_2_) at pH 7 by atomic absorption (1:10) (Thermo Fisher Scientific, Karlsruhe, Germany), whereas phosphorus (P) was determined using the Olsen method modified by extraction of sodium hydrogen carbonate at pH 8.5 using an AutoAnalyzer (1:20) (Technicon, New York, NY, USA).

### 2.3. Next-Generation Sequencing and Processing of Reads

For amplicon 16S sequencing, the DNA fragments were sequenced using a MiSeq Reagent Kit V3 (Ilumina, San Diego, CA, USA) on an Illumina MiSeq platform (Illumina, San Diego, CA, USA) for the V3–V4 variable regions of 16S rRNA, using 300 bp paired-end sequencing reads with the forward (5′ CCTACGGGNGGCWGCAG 3′) and reverse (5′ GACTACHVGGGTATCTAATCC 3′) primers. The library preparation protocol was based on the Illumina 16S rRNA Metagenomic Sequencing Library. The quality of the reads was assessed using FastQC software version 0.11.8 [29] before further analysis.

Raw sequencing reads were pre-processed using Cutadapt software version 3.4 [30] to remove adapter sequences and include all the reads with a minimum length of 200 bp. The sequences were processed using QIIME 2 software version 2021.4.0 [31] with the dada2 plugin, and reads with a maximum expected error of more than 2 (max-ee > 2) were removed during the denoising step, along with the filtering of singletons and low-abundance operational taxonomic units (OTUs). Taxonomic classification of the generated OTUs was performed using the SILVA database [32] trained for the V3–V4 region.

For whole metagenome sequencing, the DNA was first analyzed to ensure that the samples had sufficient integrity and concentration for optimal library preparation. As a standard procedure, 1 μL of gDNA from each sample was used to test integrity and purity by 1.5% agarose gel electrophoresis. Subsequently, a Qubit fluorometer was used to quantify DNA and determine whether it was sufficient to proceed; the integrity of this DNA was also verified. Thereafter, the samples were used for library preparation and sequencing. The generated DNA fragments (DNA libraries) were sequenced using an Illumina Novaseq platform (Ilumina, San Diego, CA, USA), using 150 bp paired-end sequencing reads.

### 2.4. Metabarcoding and Metagenomic Functional Analysis

From the taxonomic classification of the OTUs, data were imported into R software version 4.1.3 [33] for statistical analysis. We evaluated the dominant taxa with relative abundance using the *Phyloseq* package [34]. The relative abundance bar plots of the top phylum and family taxa were constructed using *ampvis2* [35] which is an R-package for convenient visualization and analysis of 16S rRNA amplicon data in different ways.

Diversity was assessed using the Simpson and Shannon indices for alpha diversity and the Bray Curtis index (with UPGMA cluster and NMDS representation) for beta diversity (*vegan*—*Adonis* package) [36], and the statistical differences in beta diversity between groups of samples were elucidated using permutational multivariate analysis (Permanova test). To visualize the variation in beta diversity, we used principal component analysis (PCA) to generate a graph. The resulting PCA plot was used to identify clusters of samples that correspond to the different experimental sites.

The differentially abundant taxa between groups were determined using linear discriminant analysis effect size (LEfSe analysis) [37]. For all statistical comparisons, differences were considered significant at *p* < 0.05.

The bacterial abundance was correlated with the soil and climatic parameters to estimate the impact of these variables on the microbial community of the soil and gut. Environmental variables were analyzed and filtered for correlation and multicollinearity with *corrplot* and VIF test from *usdm* R package. The filtered variables were assessed to the microbiome abundance using canonical correlation analysis (CCA) constrained to the density variable and the data was transformed initially by applying the Hellinger transformation [38].

The raw reads generated from whole metagenome sequencing were initially filtered to remove low-quality and adapter sequences. This was achieved using Trimmomatic version 0.39 [39]. The filtering criteria were set to a minimum Phred score of 20 and a minimum length of 50 bp. The filtered reads were quality-checked using FastQC version 0.11.8 [29]. Any remaining low-quality or contaminating sequences were removed using a contaminant database and Deconseq version 0.4.3 [40]. Finally, filtered and quality-checked reads were used for subsequent analyses. Raw reads generated by sequencing the whole metagenome were assembled using MegaHit software version 1.2.9 [41] pre-set with meta-sensitive k-mer parameters. Prokaryotic gene prediction of each assembled metagenome contig was carried out using Prodigal software version 2.6.3 [42] with –p meta option. eggNOG-mapper version 2.1.9 [43] was used for functional annotation of metagenome genes based on fast orthology assignments using precomputed eggNOG v5.0 clusters and phylogenies. The KEGG orthologs, CAZymes (carbohydrate-active enzymes), and taxonomic assignments of the genes were extracted using the *tidyverse* package in R [44] to generate KO and CAZyme count tables.

To perform functional enrichment analysis, we used the KEGG Mapper Reconstruct tool to group and summarize the gene count tables into pathways and gene/protein families. Differential abundances between the groups were elucidated using LEfSe analysis. Circos plots [45] were generated to visualize the distribution of the identified CAZymes across the different bacterial families present in the larval gut microbiome.

## 3. Results

### 3.1. Metadata of Amplicon 16S Sequence

Next-generation sequencing (NGS) generated 2,445,131 raw bacterial reads from all larval and soil samples (Accession Code: PRJNA933566). A total of 1,276,377 raw reads were generated from 12 larval samples. Of these, 642,033 larval sequences were obtained from the HD site (6 samples), while 634,344 larval sequences were obtained from the LD site (6 samples). A total of 1,168,754 raw reads were generated from the 12 soil samples. Of these, 594,334 sequences were obtained from the HD site (6 samples) and 574,420 sequences were obtained from the LD sites (6 samples). After quality control and denoising of the raw sequences, a total of 598,595 high-quality sequences were obtained: 380,328 from larvae and 218,267 from soil (Appendix A). The rarefaction curves of sequencing depth for both larval and soil samples reached a saturation plateau in all cases, indicating that the amount of sequencing data was sufficient and that the diversity of the samples was close to saturation, covering the species composition of the microbial community.

A total of 3278 bacterial OTUs were identified in the samples. Interestingly, the number of OTUs was similar between soil groups: 1240 and 1165 OTUs at HD and LD sites, respectively. In contrast, the number of OTUs obtained from larvae from the HD site was higher than that from larvae from the LD site, with 1472 and 1222 OTUs, respectively. For diversity analysis, counts were rarefied to a sample depth of 9659, resulting in the removal of 40 OTUs. In total, 94% and 84.6% OTUs were classified at the taxonomic level of family and genus, respectively, and only 1134 (34.6%) were classified at the species taxonomic level, which is average for metabarcoding studies with the 16S rDNA gene.

### 3.2. The Diversity and Composition of the Larval Microbiomes Differ between Insect Populations, While Soil Microbiome Remains Consistent

The gut microbiome of the larvae was assessed for microbial alpha diversity using the Shannon and Simpson indices. The gut microbiome of larvae from the HD site presented a significantly higher alpha diversity (*p* < 0.05), compared with that of larvae from the LD site (Figure 1A). However, the species richness of the gut bacterial communities did not differ significantly (Simpson index range: 0.985–0.993) between larvae collected in both sites (Appendix A). Concerning the soil microbiome, the diversity did not differ significantly between the high- and low-density sites with a Simpson index ranging from 0.972 to 0.996 (Figure 1B).

The structural composition assessed using the Bray-Curtis beta diversity index was analyzed using a principal component analysis (PCA) (Figure 2) and a cluster dendrogram (Appendix A). The composition of the soil and larval microbiomes were separated into different clusters in the dendrogram and PCA (Permanova, *p* < 0.05). The larval microbiome differed between the HD and LD sites (Permanova, *p* = 0.003). In the PCA, larval samples from the HD and LD sites formed separate cluster groups, indicating that there was a significant difference in the larval gut microbiome composition between the two sites. Regarding soil microbiome, the samples from the HD and LD sites were mixed in the dendrogram and overlapped in the PCA, indicating non-compositional differences (Permanova, *p* = 0.410).

### 3.3. Soil Microbiomes Revealed Minor Variations between Density Sites

To determine the taxonomic composition of the bacterial communities, OTUs were used to calculate the relative abundance of taxa. To avoid misclassification, we evaluated the bacterial community up to the family level, since the V1–V3 region of the 16S rRNA gene has high-level similarity between closely related taxa.

In the soils from the HD and LD sites, eight phyla represented more than 96% of the bacterial microbiota, with *Firmicutes* being dominant (24.1%), followed by *Proteobacteria* (23.2%), *Actinobacteriota* (12.3%), *Verrucomicrobiota* (9.8%), *Acidobacteriota* (9.6%), *Planctomycetota* (9.1%), *Chloroflexi* (4.6%), and *Bacteroidota* (3.3%) (Figure 3A). The most abundant families were *Bacillaceae* (18.1%) from *Firmicutes* and *Xanthobacteraceae* (11.4%) from *Proteobacteria* (Figure 3B).

However, LEfSe analysis revealed discriminating taxa (abs LDA score > 2.0) between HD and LD soils in the less abundant phyla. *Coriobacteriia* and WS2 were distinctive of HD soils, whereas *Thermoleophilia*, *Gemmatimonadota*, Pla4_lineage, and Subgroup_25 were distinctive of LD soils (Figure 3C). Soils from LD sites had 11 enriched families, with the Pla4_lineage being exclusive and representing a family-level biomarker. Soils from the HD site had six more abundant families, with WS2 and *Lachnospiraceae* as exclusive biomarkers, despite the low relative abundance they have (<0.1%) (Figure 3D).

### 3.4. Some Bacterial Families Were More Enriched in the Microbiome of Larvae from HD than from LD Sites

The gut microbiome was predominantly composed of *Firmicutes* (66.9%), *Bacteroidota* (21.4%), and *Desulfobacterota* (7.4%), which together accounted for over 95% of the relative abundance in larvae from both the HD and LD sites (Figure 4A). Within *Firmicutes*, the most abundant families were *Ruminococcaceae* (17.8%), *Lachnospiraceae* (13.4%), *Christensenellaceae* (9.7%), *Clostridiaceae* (8.1%), *Oscillospiraceae* (4.1%), *Sporomusaceae* (3.4%), and *Bacillaceae* (2.9%), whereas *Rikenellaceae* (18.3%) from *Bacteroidota*, and *Desulfovibrionaceae* (7.8%) from *Desulfobacterota* were also highly represented. Although the most abundant phyla were similar at both infestation sites, the relative abundance of families was different (Figure 4B).

LEfSe analysis revealed discriminating taxa (abs LDA score > 2.0) between larvae from both sites. The phyla *Firmicutes* and *Desulfobacterota* (>7%) and the phyla *Chloroflexi*, *Deferribacterota*, *Planctomycetota*, *Myxococcota*, and *Alphaproteobacteriota* (<1%) were distinctive of larvae from the HD site, whereas the phyla *Bacteroidota* (>20%) and BRH_c20a (<1%) were distinctive of larvae from the LD site (Figure 4C).

At the family level, we found 19 significantly discriminative taxa (LEfSe: abs LDA score > 2.0) in the gut microbiome of larvae from the two sites (Figure 4D). The families *Rikenellaceae* (22.5%) and *Dysgonomonadaceae* (2.8%) were more abundant in larvae from the LD site, whereas 15 families, led by *Desulfovibrionaceae* (9.4%) and *Clostridia* vadinBB60 group (5.1%), were more abundant in larvae from the HD site. Additionally, four families (*Nitrosomonadaceae*, *Myxococcaceae*, *Anaerofustaceae*, and *Desulfarculales*) were found exclusively in the larvae from the HD site.

### 3.5. Metagenomic Analysis Uncovers Enhanced Functional Enrichment in Larvae from the HD Site

The whole metagenome analysis of the larval gut microbiome allowed the identification of a range of 674,828 to 969,463 prokaryotic genes (predicted using Prodigal software version 2.6.3). From these, 355,460 to 517,742 genes were annotated, resulting in 13,926 distinct KEGG orthology molecular functions. In total, 267 pathways and 54 gene/protein families were identified in the gut microbiome of *P. japonica* larvae. Among these, 33 pathways and 18 gene/protein families were significantly discriminative between the microbiome of larvae from the LD and HD sites (LEfSe: abs LDA score > 2.0). Moreover, we identified 13 biomarker pathways in the microbiome of larvae from the LD site and 20 from the HD site (Figure 5A). Additionally, 5 protein families were more abundant in the microbiome of larvae from the LD site and 13 in the microbiome of larvae from the HD site (Figure 5B).

A few relevant pathways and proteins were found among the discriminative functional biomarkers, such as the lysine, valine, leucine, and isoleucine degradation pathways and the folate biosynthesis, fatty acid degradation, glutathione metabolism, bisphenol degradation, and biosynthesis of cofactors pathways, which were all enriched in larvae from the HD site. Additionally, relevant genes and proteins were enriched in larvae from the HD site, such as genes coding for cytochrome P450 and for mitochondrial biogenesis, cytokine receptor, and membrane trafficking proteins. Although larvae from the LD site had fewer enriched functional biomarkers, relevant pathways were also observed, such as those for fructose, galactose, and mannose metabolism; starch and sucrose metabolism; and two-component systems. Furthermore, genes linked to antimicrobial resistance were enriched in larvae from the LD site.

Producers of CAZymes in the microbiome have been identified to play a key role in herbivorous nutrition; thus, we checked for genes encoding these enzymes in the metagenome and for their respective producers. We identified 138 CAZyme families in the gut microbiome of the larvae. LEfSe analysis performed between larval groups from the LD and HD sites revealed that 33 CAZymes were significantly different (abs LDA score > 2.0). Only 6 CAZymes (all glycosylhydrolases) were more abundant in larvae from the LD site, whereas 27 CAZymes (mostly glycosyltransferases and some glycosylhydrolases) were enriched in larvae from the HD site (Figure 6).

To determine the producers of these CAZymes, we generated a Circos visualization plot that allowed to conclude that *Ruminococcaceae*, *Clostridiaceae*, and *Paenibacillaceae* were the most relevant producers of these enzymes in larvae from LD sites, whereas, in the microbiome of larvae from the HD site, these enzymes were produced by *Ruminococcaceae*, *Rikenellaceae*, and *Clostridiaceae* (Figure 7). CAZymes belonging to the GH5, GH9, and GT51 families were mainly produced by *Ruminococcaceae* and *Clostridiaceae* family members, with similar patterns in the microbiomes of larvae collected from both infestation sites.

### 3.6. The Core Microbiome Shared between Soil and Larvae in the HD Site, Is More Diverse and Abundant than in the LD Site

The microbiomes of the soil and larvae from the HD site shared 120 OTUs (25.5%), twice the number and relative abundance of those from the LD site (44 OTUs, 11.2%) (Figure 8A,B). These common OTUs constituted the shared microbiome, and the majority belonged to the *Bacillaceae* family (16 in the HD site and 11 in the LD site), which was the most abundant taxon in soil microbiomes. Other bacterial families shared between larvae and soils of high and low densities were *Planococcaceae* (7 and 4 OTUs, respectively), *Xanthobacteraceae* (6 and 4 OTUs, respectively), *Pirellulaceae* (6 and 3 OTUs, respectively), and *Rhizobiaceae* (5 and 2 OTUs, respectively).

To investigate the differences in the shared microbiome between the two sites, we performed a LEfSe analysis. Two OTUs belonging to *Ruminococcaceae* and one belonging to *Rikenellaceae* were enriched in the HD site, whereas one OTU belonging to UCG_010 and one belonging to *Dysgonomonadaceae* were enriched in the LD site (Figure 8C).

The gut microbiome of *P. japonica* had 49 enriched bacterial genera (LEfSe: abs LDA score > 2.0) relative to the soil microbiome (Figure 9A). Thus, comparing the microbiomes of the HD and LD sites, the most enriched genera were *Alistipes* (11% in HD and 21% in LD), *Candidatus Soleaferrea* (13% in HD and 15% in LD), *Tyzzerella* (10% in HD and 11% in LD), *Desulfovibrio* (9% in HD and 6% in LD), *Christensenellaceae* R-7 group (7% in HD and 6% in LD), *Clostridia* vadinBB60 group (5% in HD and 2% in LD), *Clostridia* UCG-014 (2% in both sites), *Clostridia* UCG-10 (2% in HD and 3% in LD), and *Dysgonomonas* (1% in HD and 3% in LD).

The soil microbiome contained 180 enriched bacterial genera relative to the larval gut microbiome (LEfSe: abs LDA score > 2.0) (Figure 9B). Thus, comparing HD and LD site soils, the most enriched genera were *Bacillus* (16% in HD and 15% in LD), *Candidatus Udaeobacter* (5% in HD and 7% in LD), *Bradyrhizobium* (3% in HD and 4% in LD), *WD2101_soil_group* (2% in both sites), *Lysinibacillus* (2% in HD and 1% in LD), and *Candidatus Xiphinematobacter* (2% in HD and 1% in LD).

### 3.7. Microbiome Biomarkers from the Two Density Sites Are Correlated with Soil Parameters

Soil and weather parameters were obtained for both density sites (Appendix A) and correlated with soil and gut microbiomes using canonical correlation analysis (CCA). The parameters were filtered to avoid the presence of autocorrelated variables in the analysis and so, OM, Mg, and Ca were considered as one variable, as well as silt and sand which presented a negative correlation. Besides, pluviometry, temperature, humidity, radiation and wind were joined as one climatic variable. Among these filtered parameters, the joined variable OM, Mg, and Ca showed the strongest correlation (*p* < 0.05) with variations in gut microbiome composition (Figure 10A). Interestingly, some OTUs that had a strong positive correlation with the parameters, which were the most extreme OTUs in the CCA plot, were also among the biomarker families of larval microbiomes from the LD site, such as *Dysgonomonadaceae*, *Clostridia* UCG-010, and *Rikenellaceae*. In contrast, those OTUs that had a negative correlation with the parameters were among the biomarker families of larval microbiomes from the HD region, such as *Clostridia vadinBB60* group and *Enterococcaceae* RF39.

Concerning soil microbiomes, also the joined variable Ca, Mg and OM showed the strongest correlation with microbial diversity (Figure 10B). In addition, some OTUs that had a negative correlation with the parameters were among the biomarker families from the soil of the HD site, such as WS2, *Rhodomicrobiaceae* and *Nocardiaceae*. Additionally, some OTUs from biomarker families, such as *Streptomycetaceae, Mycobacteriaceae and Gemmatimonadaceae*, from the LD site were highly correlated with Ca, Mg, and OM.

## 4. Discussion

### 4.1. The Gut Microbiome of Larvae from HD Sites Is More Favorable to Insect than That of Larvae from LD Sites

The gut microbiome is known to influence the biology of many organisms, including insects where it influences their survival and fitness. The prevalent bacterial families in the gut of *P. japonica* larvae were *Rikenellaceae*, *Ruminococcaceae*, and *Lachnospiraceae*, which are bacterial families common in several insects. *Ruminococcaceae* and *Rikenellaceae* were found in the gut microbiome of other insects, such as termites, beetles, and cockroaches. These are important bacterial families that are involved in the digestion of wood and other plant materials [46,47,48,49]. They also help insects to obtain energy and regulate host intestinal physiology by producing short-chain fatty acids [20,50].

However, what our study points out is that even though the prevalent families were the same in both microbiomes, important bacterial biomarkers were evident when we focus on the less dominant families. *Desulfovibrionaceae* and *Clostridia* vadinBB60 should be considered important biomarkers in the gut microbiome of larvae from the HD site whereas in the microbiome of larvae from the LD site should be *Rikenellaceae*, *Dysgonomonadaceae* and *Clostridia* UCG_010. *Dysgonomonadaceae* is an important family, which is also found in the gut of insects that feed on lignocellulose-rich food as well as in the systems for lignocellulose biomass conversion into biogas. This shows that members of this family are essential for breaking down plant cell walls. For example, *Dysgonomonas gadei*, found in the gut of *Pachnoda marginata*, is a facultative anaerobe that can transform a variety of carbohydrates, such as cellobiose, fructose, lactose, starch, sucrose, and xylose, into various acids [20]. Another significant biomarker in larvae is the UCG_010 group within the *Clostridia* class, which exhibits a remarkable ability for carbohydrate assimilation by utilization of xylan and cellulose [51,52,53].

Despite this, larvae from the HD site have a higher diversity of enriched bacterial families related to plant digestion and nutrient acquisition. *Desulfovibrionaceae* is an important family of bacteria, extensively found in the gut of many insects, such as *Melolontha melolontha* and *Cephalodesmius* sp. [54,55], termites [56,57], and cockroaches [58]. These bacteria are sulfate-reducing, which are also facultative oxygen and hydrogen consumers and acetate producers [20]. Apparently, they actively contribute to lowering the sulfate concentration between the midgut and hindgut of *M. melolontha*, reducing the toxicity (by detoxification) of this harmful compound can have on insects [59]. We hypothesize that this bacterial family is maintaining the gut microenvironment of JB grubs thus supporting their development.

Another interesting biomarker found in larvae from the HD site was the vadinBB60 group of the *Clostridia* class. This taxonomic group has a cellulosome-like structure, which is a large protein complex responsible for the degradation of plant cell walls. This enzymatic complex allows bacteria to efficiently degrade cellulose and other plant polysaccharides [60]. *Clostridia* is considered a symbiont of higher termites and was found to be localized in the gut-mixed segment of the termite *Nasutitermes takasagoensis* [16]. Furthermore, the fact that many taxonomic groups of *Clostridia* have been found to degrade polysaccharides into acetone, alcohol, lactate, CO_2_, and hydrogen [51,52,53,61,62], and that other groups can ferment nitrogenous or lipidic compounds [63,64], suggesting that this bacterial group significantly contributes to insect nutritional requirements.

Equally important are the exclusive taxa detected in the gut of HD site larvae: *Nitrosomonadaceae* are known to participate in the nitrogen cycle and the oxidation of ammonia to nitrite [65] and *Desulfarculales* participate in the conversion of sulfate to hydrogen sulfide [66]. These taxa may be crucial for the growth of gut bacteria and for the overall health and nutrition of the insect host and in fact, they are prevalent in the microbiome of the larval gut of *P. japonica* in the populations from the high-density site.

### 4.2. Metagenomic Analysis Uncovers Enhanced Functional Enrichment in the Larvae from the HD Site

The metagenomic analysis of the *P. japonica* gut microbiome showed enhanced functional enrichment in the HD site larvae compared to that of the LD site. The larvae from the HD site had a higher abundance of biomarker pathways, genes/protein families, and CAZymes than did the larvae from the LD site. These findings suggest that the gut microbiome of the larvae from the HD site is more enriched and has a greater potential for nutrient utilization, improved plant material degradation, detoxification, and enhanced immune response than the gut microbiome of the larvae from the LD site.

The different pathways found to be enriched in larvae from the HD site indicated a better capability to break down several amino acids (lysine, valine, leucine, and isoleucine) as a source of energy [67], produce vitamins (B-vitamin) [68,69], biosynthesize cofactors for enzymes, degrade fatty acids, degrade biophenols, and produce glutathione (which is an antioxidant that protects the host) [70]. In contrast, LD larvae with less enriched features displayed different enriched pathways associated with sugar metabolism (fructose, mannose, pentose, galactose, starch, and sucrose metabolism) and a pathway related to the adaptation of bacteria to changes in their environment (two-component system). This pathway enables bacteria to sense, respond, and adapt to a wide range of environments and stressors. This observation could indicate the presence of bacteria with greater resistance and higher tolerance to drugs (e.g., antibiotics, pesticides) or chemicals existing in the soil [71]. The analysis of the genes and proteins that were enriched in larvae from the HD site also showed that they could have a better capacity for detoxification (cytochrome P450) [72,73], improved energy production and cellular metabolism (mitochondrial biogenesis) [74], and better regulation of the immune system (bacterial cytokine receptors) [75]. In fact, the gut microbiome of larvae from the HD site presented more enriched functional features than larvae from the LD site, which suggests that it can carry out a variety of metabolic tasks that are crucial for insect survival and well-being. Conversely, larvae from the LD site showed enrichment in only a few genes and proteins, which once again appear to be associated with antimicrobial resistance (antimicrobial resistance genes) and the sensing and response mechanisms to environmental changes (two-component system).

The presence of a higher number of CAZymes in the microbiome of larvae from the HD site indicated a potential advantage in comparison with those from the LD site, in terms of herbivorous nutrition. The GH5 and GH9 CAZyme families, mostly produced by *Ruminococcaceae* and *Clostridiaceae* families in both JB populations, are important for the digestion of carbohydrates in insect guts. The complex carbohydrate xylan, which constitutes a significant portion of plant cell walls, is broken down by members of the enzymatic family GH5, known as xylanases [76]. Other members of this family, also referred to as α-L-arabinofuranosidases, are enzymes that degrade arabinoxylan, a complex carbohydrate that also constitutes a significant portion of plant cell walls [77]. In addition, the GH9 family includes β-glucanases that degrade β-glucans, which are polysaccharides found in plant cell walls and fungi [78]. Similar to xylanases and α-L-arabinofuranosidases, β-glucanases support insect digestion and nutrient extraction from OM. Overall, the CAZymes GH5 and GH9 play important roles in the digestion of plant materials by insects, allowing them to obtain nutrients from their diet and support their growth and development.

Interestingly, the comparison between the microbiomes of larvae from the HD and LD sites showed that, in the former, more CAZymes were being produced by members of the *Rikenellaceae* family. These CAZymes included GH65, GT2, GT3, and GT51 family members. Enzymes of the GH65 family are known to breakdown several disaccharides, such as trehalose, maltose, cellobiose, isomaltose, sucrose, and lactose [79], and those of the GT2, GT3, and GT51 families are glycosyltransferases that catalyze the conjugation of various biomolecules with sugars to produce different inactive glycosides. These GT families, which were enriched in the larval microbiome from the HD site, play an essential role in the detoxification of xenobiotics and allelochemicals found in plants as well as in providing insects with a mechanism for pesticide cross-resistance [80].

### 4.3. Larvae from the HD Site Share with Soil a Higher Number of Bacteria than Those from LD Site

Another relevant finding resulting from the comparison of microbiomes in the gut from HD and LD is that larvae from the HD site shared a higher number of OTUs with their soil than the populations from the LD site. This data is in accordance with Huang and Zhang that showed the composition and diversity of the dominant bacterial groups in *Holotrichia parallela* larvae varied significantly among ten populations from different locations [23]. Likewise, the gut microbiome of the grass grub of *Costelytra zealandica* was reported to have community variations between samples that were influenced by external factors, such as the geographic location of the population and the insects’ diet [49].

The most enriched taxa in larvae relative to the soil are all members of bacterial families known as fiber-digesting bacteria [17,23,81]. Chouaia et al. and Avila-Arias et al. found that the same bacterial families were enriched in the midgut and hindgut of JB larvae relative to the soil [22,46]. The shared microbial taxa with the soil may indicate specific horizontal transmission, which allows better assimilation of the local microbiome and adaptation to the environment and diet of the region. To be noticed that the observed differences in beta composition between the larval microbiome from the HD and LD sites could not be fully attributed to horizontal transmission because the soil microbiome was practically equal. Therefore, we assume that vertical transmission should play a key role in the microbiome of *P. japonica* larvae. Chouaia et al. compared the microbiome between soil and larvae, pupae, and adults of *P. japonica* and found that larvae share more microbiome components with soil than pupae and adults. In addition, the authors reported a small core bacterial community (89 OTUs) that was common throughout all life stages, proving the existence of vertical transmission [46].

### 4.4. There Is a Correlation between Microbiome Biomarkers and Soil Parameters

In this study, we also found the potential hint of the correlation between three soil abiotic factors (OM, Mg, and Ca) and the variations observed between the gut microbiomes of larvae from the HD and LD sites. For example, a positive correlation of these factors was observed with abundant OTUs belonging to the biomarker families of LD larvae, such as *Dysgonomonadaceae*, *Clostridia* UCG-010, and *Rikenellaceae*. On the other hand, abundant OTUs with a negative correlation to soil parameters belonged to the biomarker families of HD larvae, such as *Clostridia* vadinBB60 group and the *Enterococcaceae* RF39. These results indicated that soil parameters may have an influence on the gut microbiota of the larvae and may help the insects adapt to different ecological circumstances.

Avila-Arias et al. investigated the effects of abiotic factors on the gut microbiome of JB larvae. They found that factors such as cation exchange capacity, OM content, water-holding capacity, and texture of the soil were moderately correlated with the composition of prokaryotic microbes in the gut, particularly in the midgut. Similar to our data, these results also suggest that the microbial communities present in JB guts are partly shaped by their adaptation to local soil conditions [22].

## 5. Conclusions

This study showed that the gut microbiome of *P. japonica* larvae differs between high- and low-density populations. We could establish clear biomarkers in these microbiomes, particularly in the nondominant phyla of bacteria. In addition, the microbiome in insects from high population rates has enriched functional pathways linked to plant digestion, detoxication, and immune defenses. Furthermore, we observed that soil parameters (e.g., organic matter, Ca, and Mg) correlate negatively with bacterial biomarkers of *P. japonica* larvae from high densities. Altogether, these findings suggest that gut microbiome could be a target in a strategy to prevent pest dissemination and growth, as indicated by others [15,82]. We assume that the establishment of specific microbiome biomarkers in *P. japonica* could serve as a valuable tool to assess the risk of population spread and expansion in various regions. The findings of this study indicate a strong association between insect microbiomes and soil parameters. As a result, a new hypothesis emerges, proposing that the manipulation of soil abiotic factors could potentially influence the microbiome of JB larvae, resulting in reduced insect fitness and limited establishment in new areas. Therefore, this important research area could provide further knowledge on the intricate interactions between insects, microbes, and the environment.

## Figures and Tables

**Figure 1 microorganisms-11-01972-f001:**
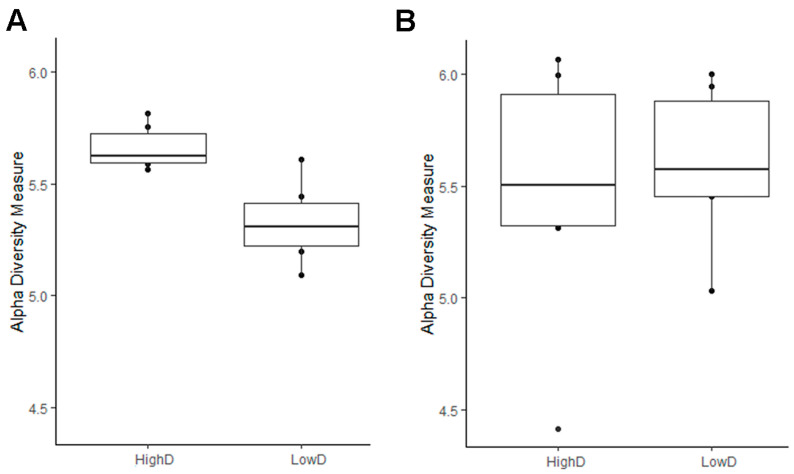
Boxplots representing the α-diversity index of Shannon from (**A**) larvae and (**B**) soil microbiomes from high- (HD) and low- density (LD) sites.

**Figure 2 microorganisms-11-01972-f002:**
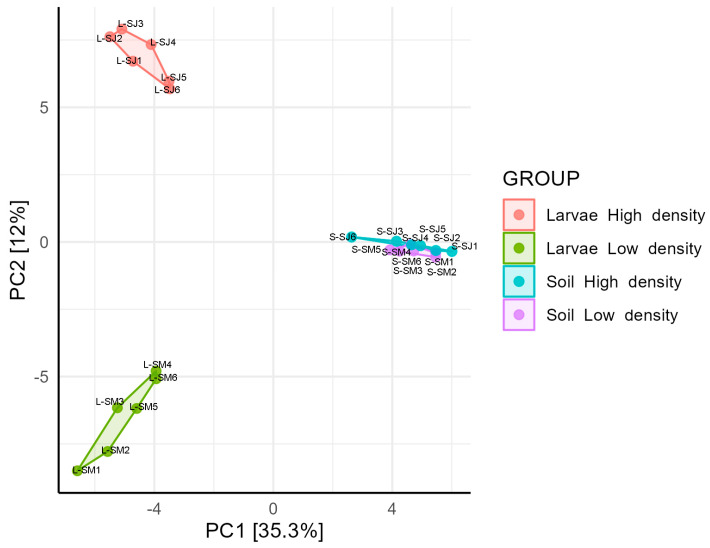
Principal components analysis (PCA) showing taxa structural composition of soil and larvae microbiomes sampled in high- and low-density sites. Data were log-transformed, and distances measured by Bray-Curtis distances.

**Figure 3 microorganisms-11-01972-f003:**
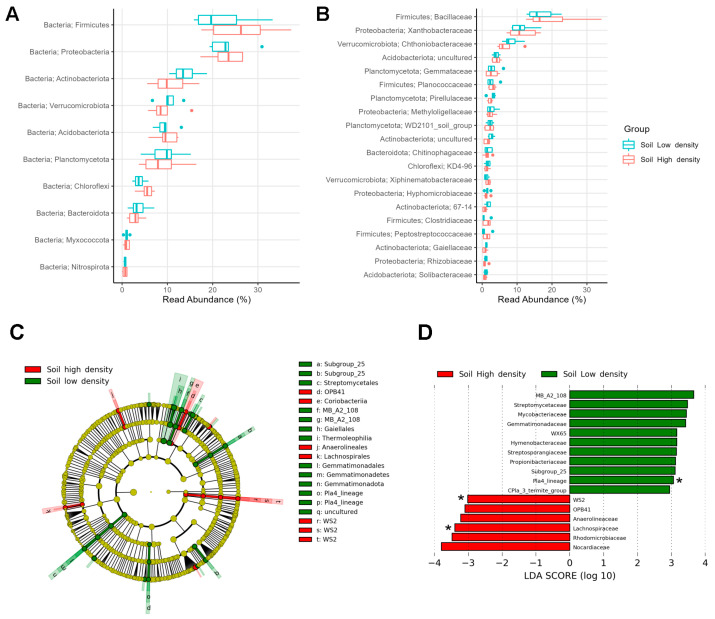
Relative abundance of top taxa found in the soil microbiomes of high- and low-density sites: (**A**) Top 10 phyla and (**B**) top 20 families. Biomarkers in soil microbiomes: (**C**) Dendrogram and (**D**) bar plot showing discriminative bacterial phyla and families between infestation sites. Asterisks (*) depict exclusive families.

**Figure 4 microorganisms-11-01972-f004:**
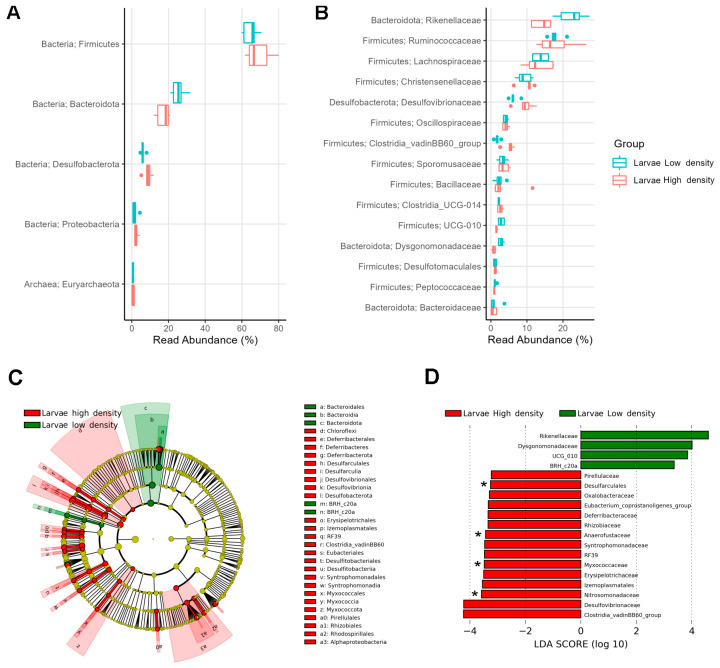
Relative abundance of top taxa found in the larval gut microbiome: (**A**) Relative abundance of top 5 phyla and (**B**) top 15 families found in larval group samples from high- and low-density sites. Biomarkers in the gut microbiomes of larvae: (**C**) Dendrogram and (**D**) bar plot showing discriminative phyla and families between infestation sites. Asterisks (*) depict exclusive families.

**Figure 5 microorganisms-11-01972-f005:**
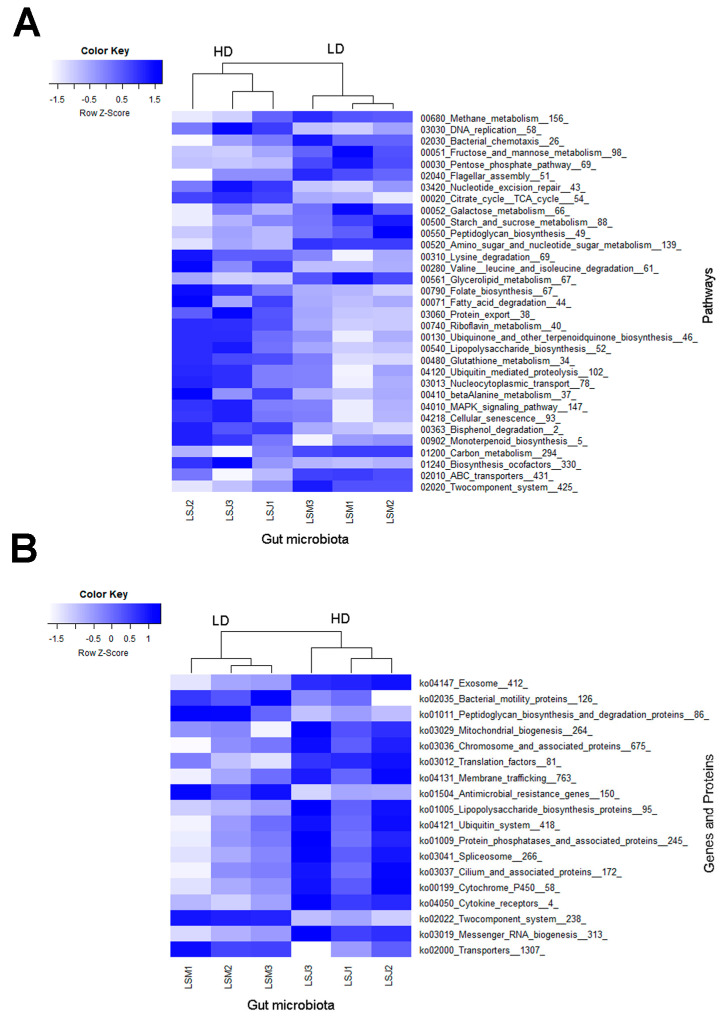
Heatmaps showing differences in metagenomic (**A**) KEGG pathway (level 3) and (**B**) gene/protein families between the gut microbiomes of larvae from high- and low-density sites, analyzed by LEfSe (LDA > 2.0, *p* < 0.05). Samples nomenclatures are defined in Appendix A.

**Figure 6 microorganisms-11-01972-f006:**
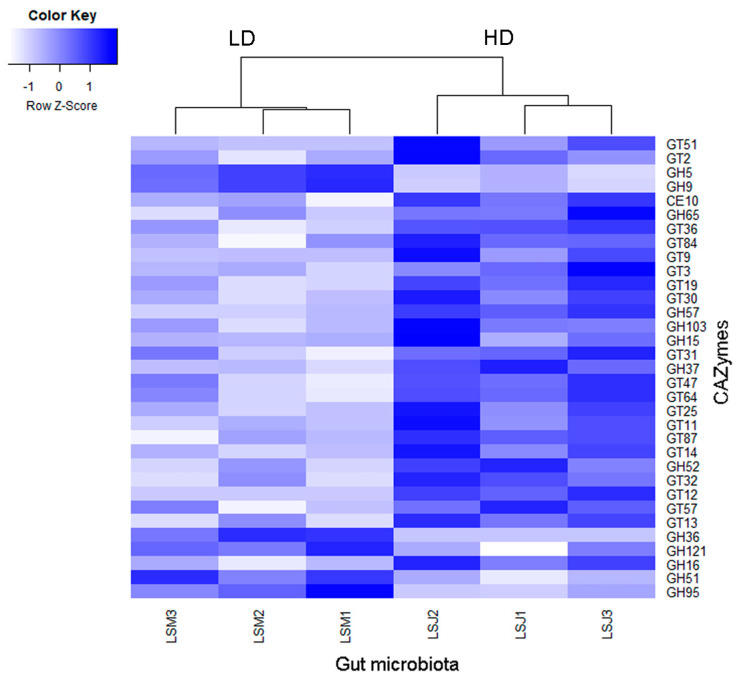
Heatmap showing differences in the metagenomic CAZymes between the gut microbiomes of larvae from high- and low-density sites, analyzed by LEfSe (LDA > 2.0, *p* < 0.05). Samples nomenclatures are defined in Appendix A.

**Figure 7 microorganisms-11-01972-f007:**
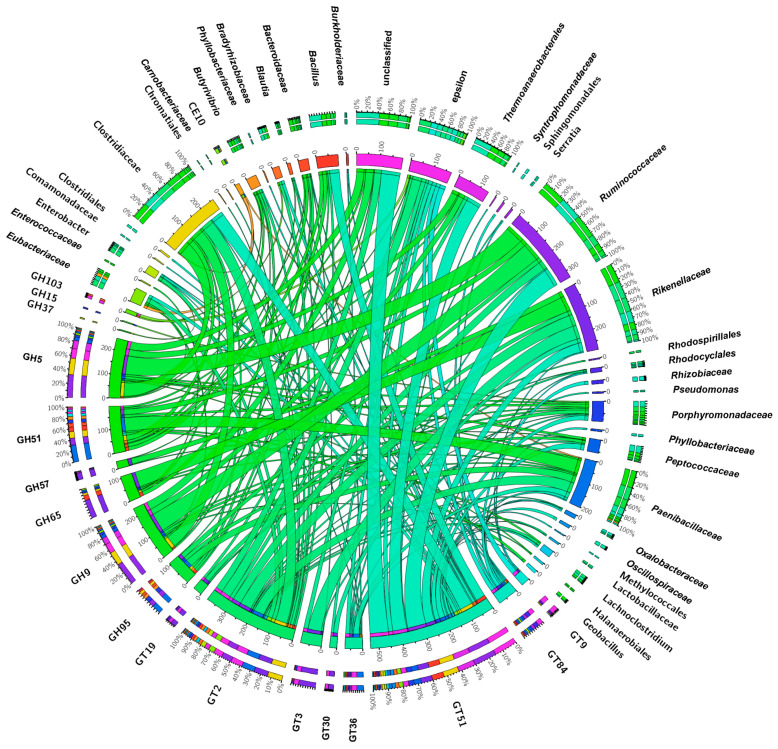
Identified CAZymes and associated bacterial families in larval gut microbiomes. The Circos plot illustrates the CAZyme classes found and their distribution across the bacterial families in larvae from the HD site.

**Figure 8 microorganisms-11-01972-f008:**
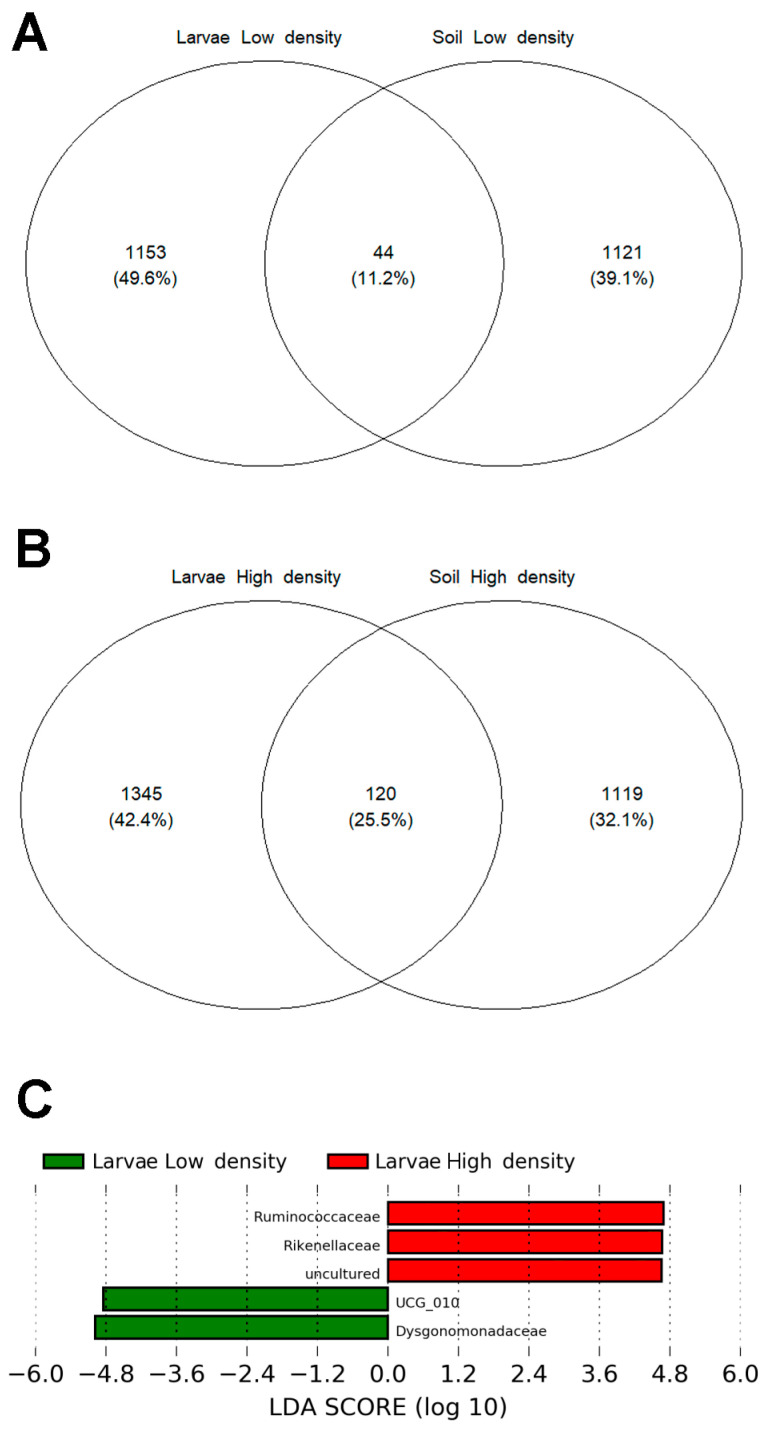
Venn diagram showing core microbiome (OTUs) shared between larvae and soil in (**A**) low- and (**B**) high-density sites (LD and HD, respectively). (**C**) Differentially abundant bacterial families between shared microbiomes in HD and LD sites: 2 OTUs belonging to *Ruminococcaceae* and 1 OTU from *Rikenellaceae* were enriched in the HD site, whereas 1 OTU belonging to UCG_010 and 1 OTU from *Dysgonomonadaceae* were enriched in the LD site.

**Figure 9 microorganisms-11-01972-f009:**
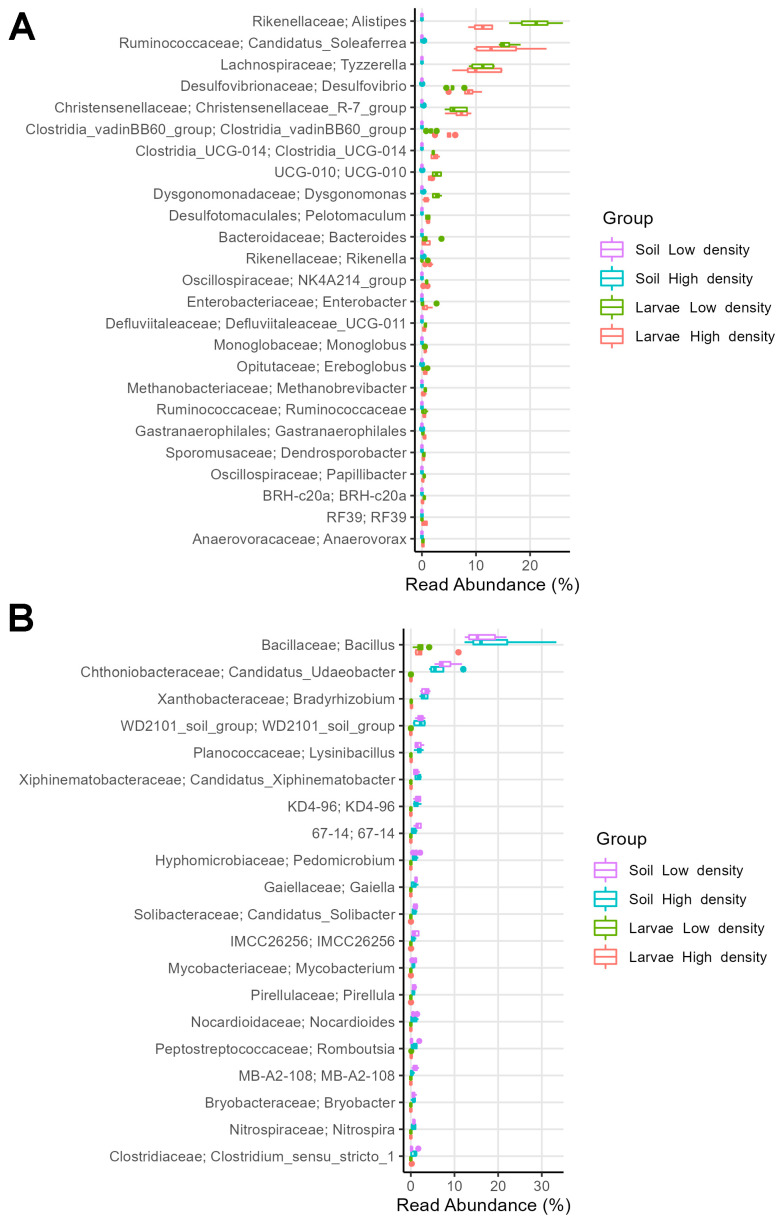
Relative abundance of (**A**) top 25 larval biomarkers (bacterial genera) relative to soil microbiomes and of (**B**) top 10 soil biomarkers (bacterial genera) relative to larval microbiomes from the HD and LD sites, determined by LEfSe analysis between soil and larval microbiomes.

**Figure 10 microorganisms-11-01972-f010:**
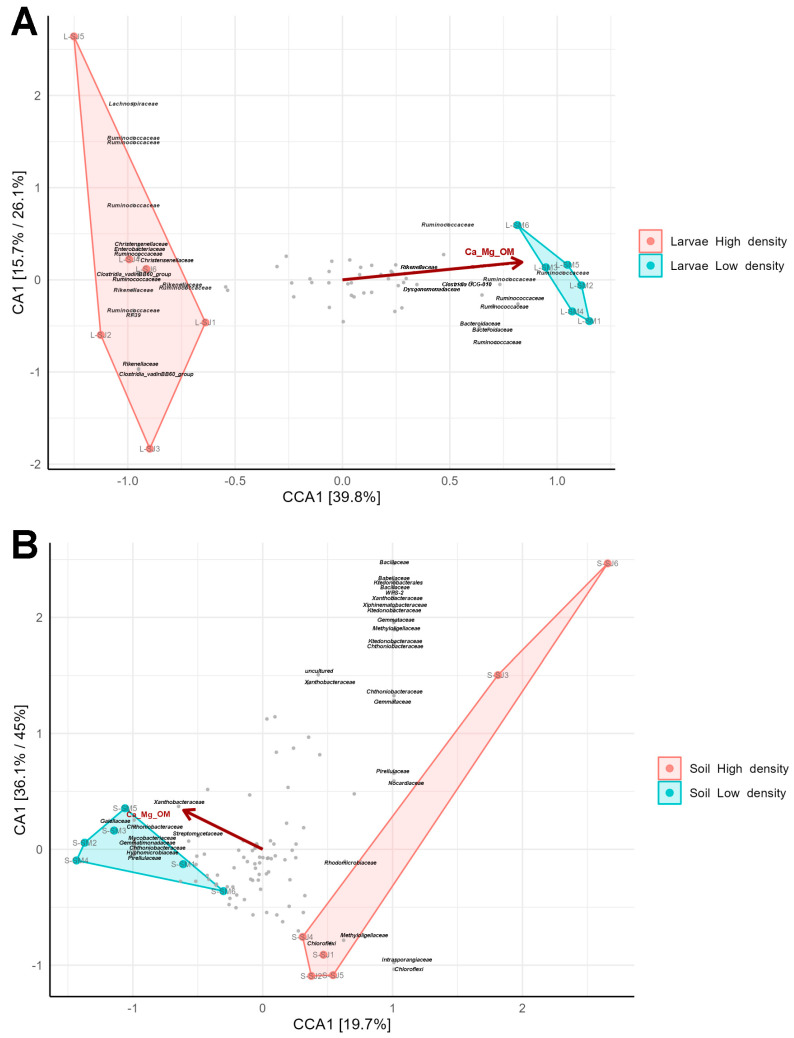
The Soil and larval microbiome correlation with abiotic factors. (**A**) Canonical Correspondence Analysis (CCA) of 65 OTUs and 12 samples constrained to the “density” variable showing the correlation between the bacterial OTUs of larvae and different soil properties (pH, OM, P, K, Ca, Mg, silt, sand, and clay). Prior to the analysis, OTU’s that are not present in more than 0.8% relative abundance in any sample have been removed. (**B**) Canonical Correspondence Analysis (CCA) of 108 OTUs and 12 samples constrained to the “density” variable showing the correlation between the bacterial OTUs of soil and different soil properties. Prior to the analysis, OTU’s that are not present in more than 0.6% relative abundance in any sample have been removed. The relative contribution (eigenvalue) of *x*- and *y*- axis data to the total inertia in the data and the constrained space, respectively, are indicated as percentages in the axes legends. The vectors represent the mean direction and strength of correlation of the different parameters measured (*p* < 0.05).

## Data Availability

The datasets of BioProject PRJNA933566 for this study can be found in the NCBI repository SRA.

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
