# Peer review of "Exploring Gut Microbiome Variations between Popillia japonica Populations of Azores"

_microorganisms, 2023, doi:10.3390/microorganisms11081972_

Round 1

Reviewer 1 Report

Article
Exploring gut microbiome variations between Popillia japonica populations of Azores

The authors worked on gut microbiome populations.
Popillia japonica (Coleoptera: Scarabaeidae), is an emerging invasive pest in Europe and America
Overall English is good,
The abstract and Title seem ok
The article is organized and structured very well with a good number of figures.
No tables were found though.
The figures were extremely good and informative, maybe more can be interpreted from the figures which can be included in the
results or discussion section.
Findings suggested that the microbiome must be one drive for the increase of P. japonica populations, thus providing a check
point in the establishment and spread of this pest.
The study also found that the gut microbiome of P. japonica larvae differs between high- and low-density populations.

This article can be accepted for publication.

With Regards,

Author Response

Response to the Reviewers’ Comments on the Manuscript entitled “Exploring gut microbiome variations between Popillia japonica populations of Azores” submitted to Microorganisms (MDPI).

We would like to thank the Reviewers for carefully reading our paper. The valuable comments and criticisms from all Reviewers have allowed us to improve our manuscript. We are very encouraged by the Reviewers' recognition of our research in terms of methodology and results presented in the manuscript. In the following, we include a point-by-point response to the questions and comments of each Reviewer. With this resubmission, we also attach a revised manuscript that includes the updated text to address the Reviewers’ concerns. We do hope that the Reviewers and editor appreciate our significant efforts in the improvement of the manuscript and consider it for publication.

Author's Notes to Reviewer 1:

We appreciate your objective assessment of the manuscript. We are encouraged by your acknowledgment of our research in terms of the approach and results presented in the manuscript.

Reviewer 2 Report

The paper authored by Frias J et al. analyzed the microbiome of the larvae and soil samples from different sites with high and low infestation rates of Popillia japonica populations. They found that the gut microbiome of larvae differs between high- and low-density populations, and the microbiome in insects from high population rates had enriched functional pathways linked to plant digestion, detoxication, and immune defenses. The experimental design of this paper is reasonable, and the result analysis is comprehensive. And, I only have some minor comments.

1. The alpha and beta diversity of the soil microbiome was similar between the high-density and low-density sites. So what is the initial origin of different biomarkers in the gut of larvae of the high-density and low-density regions? And, what is significance of establishing different biomarkers in Popillia japonica with different population density? I suggest the author to discuss.

2. The special biomarkers in the larvae from the high-density region linked to plant digestion, nutrient acquisition, and detoxification. I suggest that the author should not only analyze the weather and soil parameters, but also analyze the plant species and planting area in different density population sites, as well as the corresponding use of pest control agents.

3. What are the practical suggestions for the control of Popillia japonica baesd on the findings of this paper? I suggest the author to discuss.

Author Response

Response to the Reviewers’ Comments on the Manuscript entitled “Exploring gut microbiome variations between Popillia japonica populations of Azores” submitted to Microorganisms (MDPI).

We would like to thank the Reviewers for carefully reading our paper. The valuable comments and criticisms from all Reviewers have allowed us to improve our manuscript. We are very encouraged by the Reviewers' recognition of our research in terms of methodology and results presented in the manuscript. In the following, we include a point-by-point response to the questions and comments of each Reviewer. With this resubmission, we also attach a revised manuscript that includes the updated text to address the Reviewers’ concerns. We do hope that the Reviewers and editor appreciate our significant efforts in the improvement of the manuscript and consider it for publication.

Author's Notes to Reviewer 2:

Thank you for the review of the manuscript and for the acknowledgment of our research in terms of experimental design and results analysis. Your valuable comments and criticisms have allowed us to improve our manuscript.

1. “The alpha and beta diversity of the soil microbiome was similar between the high-density and low-density sites. So what is the initial origin of different biomarkers in the gut of larvae of the high-density and low-density regions? And, what is significance of establishing different biomarkers in Popillia japonica with different population density? I suggest the author to discuss.”

In fact, our data do not give any information about the origin of different biomarkers in the gut of larvae since there were no differences in alpha and beta diversity of soil microbiome. However other authors emphasis that the differences in microbiome may be due to vertical transmission as we discuss in the final of section 4.3. The origin of different biomarkers between larval microbiome from the HD and LD sites could be explained by vertical transmission of microbiome from isolated populations in geographically distinct islands and soil types, conjugated with a selective effect exerted by the microbiome and by the insect.

We added this last statement to the conclusion part of the manuscript showing the significance/importance of the presented data for the characterization of different P. japonica populations concerning their microbiome.

“We assume that the establishment of specific microbiome biomarkers in P. japonica could serve as a valuable tool to assess the risk of population spread and expansion in various regions.”

  1. “The special biomarkers in the larvae from the high-density region linked to plant digestion, nutrient acquisition, and detoxification. I suggest that the author should not only analyze the weather and soil parameters, but also analyze the plant species and planting area in different density population sites, as well as the corresponding use of pest control agents.”

Thank you for the valuable suggestion. We have now incorporated information about the flora composition of pastures and the surrounding areas in section 2.1 to highlight their similarities in both islands studied. Furthermore, we included in the introduction of the manuscript (lines 45-48), the historical use of various pest control agents in an attempt to control the P. japonica populations. Unfortunately, these control measures have proven ineffective in reducing the expansion of the pest's populations.

  1. “What are the practical suggestions for the control of Popillia japonica baesd on the findings of this paper? I suggest the author to discuss.”

Thanks for pointing this out. A practical approach to control P. japonica could involve manipulating soil parameters, particularly those that exhibit negative correlations with biomarkers found in HD larvae. By targeting these parameters, it may be possible to intervene in the insect microbiome and consequently influence their fitness. we have provided an explanation in the conclusion of the manuscript regarding this potential practical suggestion for P. japonica control based on our research findings. We are now carrying out new experiments to explore more practical and applied measures involving the microbiome findings.

Reviewer 3 Report

The manuscript by Frias et al. provides evidence for a connection between the gut bacterial composition of an invasive pest, i.e. the Japanese beetle P. japonica, and the abundance of the infesting populations, taking advantage of the strict geographical separation offered by the islands ecosystems. Along with confirming once again the important relation between the soil microbiota and that of soil dwelling insects, the paper underlines the relevant role played by the microbiota in modulating the invasive potential of an insect species. The methodological design is correctly conducted and presented. Some suggestions are listed below, I believe they would be helpful in improving the manuscript.

Materials and methods

When describing the selected location for samplings, the authors explain why the considered the chosen areas of 40 km2 as comparable as for the ecosystem (altitude, land use). Are these area also characterized by a comparable availability of host plants for the adults? This may be relevant since the whole paper is based on examining the difference at the soil/larvae interface that could have a role in determining the invasion success, but if other differences occur that hamper the successful development (and hence fecundity) of adults, this may bias the following observations.

Discussion

Presently the discussion is mostly focused on the traits related with HD sites; I understand they are of major interest but I suggest to expand also the discussion about the taxa that have been reported as markers of LD sites. For example, in paragraph 4.1 only the HD biomarker are described, and the LD ones should be added. Also the discussion of metagenomic analysis should include some comments on the LD-associated bacteria, as some gene groups/pathways were mostly present in this group.

Finally, I believe that the authors omitted commenting a very important implication of the results obtained in this work. Indeed, the presented finding show that the soil parameters may drive the bacterial selection in the gut of JB larvae, and this could be a proxy for the soil suitability for the establishment of HD populations.

Author Response

Response to the Reviewers’ Comments on the Manuscript entitled “Exploring gut microbiome variations between Popillia japonica populations of Azores” submitted to Microorganisms (MDPI).

We would like to thank the Reviewers for carefully reading our paper. The valuable comments and criticisms from all Reviewers have allowed us to improve our manuscript. We are very encouraged by the Reviewers' recognition of our research in terms of methodology and results presented in the manuscript. In the following, we include a point-by-point response to the questions and comments of each Reviewer. With this resubmission, we also attach a revised manuscript that includes the updated text to address the Reviewers’ concerns. We do hope that the Reviewers and editor appreciate our significant efforts in the improvement of the manuscript and consider it for publication.

Author's Notes to Reviewer 3:

Thank you for assessing our manuscript. We appreciate all the comments and used the suggestions to improve the manuscript.

  1. “When describing the selected location for samplings, the authors explain why the considered the chosen areas of 40 km2 as comparable as for the ecosystem (altitude, land use). Are these area also characterized by a comparable availability of host plants for the adults? This may be relevant since the whole paper is based on examining the difference at the soil/larvae interface that could have a role in determining the invasion success, but if other differences occur that hamper the successful development (and hence fecundity) of adults, this may bias the following observations.”

Thanks for pointing this out. We have now included a characterization of the flora composition of pastures and the surrounding areas in section 2.1 to highlight their similarities in both islands studied. Notably, both islands exhibit similar availability of host plants, with the surrounding vegetation of pastures predominantly composed of Rubus spp. (blackberry, raspberry), which we observed to be the most affected plant by P. japonica adults.

  1. “Presently the discussion is mostly focused on the traits related with HD sites; I understand they are of major interest but I suggest to expand also the discussion about the taxa that have been reported as markers of LD sites. For example, in paragraph 4.1 only the HD biomarker are described, and the LD ones should be added. Also the discussion of metagenomic analysis should include some comments on the LD-associated bacteria, as some gene groups/pathways were mostly present in this group.”

Indeed, our primary focus was on features related to larvae from HD sites. However, in response to your feedback, we have expanded the discussion in section 4.1 to include additional information about the biomarkers found in LD larvae gut, such as Dysgonomonadaceae and Clostridia UCG_010, which are also vital for nutrient utilization (lines 471-480). Moreover, we have addressed pathways and genes enriched in LD larvae that appear to be associated with sugar metabolism, resistant genes, and sensing/response mechanisms in section 4.2 (lines 521-528 and 534-538)

.3. “Finally, I believe that the authors omitted commenting a very important implication of the results obtained in this work. Indeed, the presented finding show that the soil parameters may drive the bacterial selection in the gut of JB larvae, and this could be a proxy for the soil suitability for the establishment of HD populations”

Thank you for your valuable input. We completely agree with your perspective. As a practical approach to control P. japonica, manipulating soil parameters, especially those negatively correlated with biomarkers found in HD larvae, could be a promising strategy. By targeting these parameters, we may have the opportunity to influence the insect microbiome and subsequently impact their fitness. In our conclusion, we have now elaborated on this potential practical suggestion for P. japonica control, based on the findings from our research. However, it is important to note that this idea is currently a hypothesis, and further studies are required to establish if these abiotic factors indeed play a direct role in the insect microbiome and consequently the establishment of P. japonica populations.